# Toward Evaluating Critical Factors of Extubation Outcome with XCSR-Generated Rules

**DOI:** 10.3390/bioengineering9110701

**Published:** 2022-11-17

**Authors:** Po-Hsun Huang, Lian-Yu Chen, Wei-Chan Chung, Chau-Chyun Sheu, Tzu-Chien Hsiao, Jong-Rung Tsai

**Affiliations:** 1Institute of Computer Science and Engineering, College of Computer Science, National Yang Ming Chiao Tung University, Hsinchu 30010, Taiwan; 2Division of Respiratory Therapy, Kaohsiung Medical University Hospital, Kaohsiung 80708, Taiwan; 3Division of Pulmonary and Critical Care Medicine, Kaohsiung Medical University Hospital, Kaohsiung 80708, Taiwan; 4School of Medicine, College of Medicine, Kaohsiung Medical University, Kaohsiung 80708, Taiwan; 5Department of Computer Science, College of Computer Science, National Yang Ming Chiao Tung University, Hsinchu 30010, Taiwan; 6Institute of Biomedical Engineering, National Yang Ming Chiao Tung University, Hsinchu 30010, Taiwan; 7Department of Internal Medicine, Kaohsiung Municipal Cijin Hospital, Kaohsiung 80544, Taiwan; 8Division of Respiratory Therapy, College of Medicine, Kaohsiung Medical University, Kaohsiung 80708, Taiwan

**Keywords:** mechanical ventilation, spontaneous breathing trial, successful extubation, extended classifier system

## Abstract

Predicting the correct timing for extubation is pivotal for critically ill patients with mechanical ventilation support. Evidence suggests that extubation failure occurs in approximately 15–20% of patients, despite their passing of the extubation evaluation, necessitating reintubation. For critically ill patients, reintubation invariably increases mortality risk and medical costs. The numerous parameters that have been proposed for extubation decision-making, which constitute the key predictors of successful extubation, remains unclear. In this study, an extended classifier system capable of processing real-value inputs was proposed to select features of successful extubation. In total, 40 features linked to clinical information and variables acquired during spontaneous breathing trial (SBT) were used as the environmental inputs. According to the number of “don’t care” rules in a population set, Prob_usage_, the probability of the feature not being classified as above rules, can be calculated. A total of 228 subjects’ results showed that Prob_usage_ was higher than 90% for minute ventilation at the 1st, 30th, 60th, and 90th minutes; respiratory rate at the 90th minute; and body weight, indicating that the variance in respiratory parameters during an SBT are critical predictors of successful extubation. The present XCSR model is useful to evaluate critical factors of extubation outcomes. Additionally, the current findings suggest that SBT duration should exceed 90 min, and that clinicians should consider the variance in respiratory variables during an SBT before making extubation decisions.

## 1. Introduction

In the intensive care unit (ICU), intubation with mechanical ventilators necessarily saves the lives of critically ill patients immediately fighting for their lives and gaining extra time to treat and recover. However, clinical reports provide evidence that prolonged mechanical ventilation can lead to the impairment of diaphragmatic or lung function [1,2,3]. Similarly, respiratory failure requiring reintubation occurs in approximately 15–20% of patients after extubation [4]. Delayed extubation or reintubation tends to increase mortality risk in critically ill patients. Therefore, it is critical to accurately determine whether patients still require mechanical ventilation support. The timing of weaning from mechanical ventilation also constitutes a crucial concern.

Patients who are intubated and undergoing machine ventilation support require frequent physician visits and checkups. After the stabilization of original cause, there is a procedure for determining the patient’s health status called the ventilator weaning protocol. The first step is to observe the patient’s daily assessment of readiness to wean, and to meet and check the weaning profiles, including the rapid shallow breathing index (RSBI), maximal inspiratory pressure (Pimax), cuff leak test, etc. [5,6,7,8]. The second step was to conduct a spontaneous breathing trial (SBT) with 120 min of low pressure support ventilation. The third step of airway protection and cough function assessment was performed. After all of them have passed, the decision to turn off mechanical ventilation and remove the endotracheal tube is made by a trained and certified physician. The strictly refined process still produces 15~20% extubation failure rate; therefore, many studies have introduced multivariate analysis, artificial intelligence, and other methods to find the appropriate factors. During SBT, respiratory distress may occur [9] and changes in the respiratory variables may also be associated with successful extubation [10]. It is, however, still not possible to completely exclude the possibility of failed extubation. The search for other key variables, the development of novel features, and the establishment of accurate predictive models for extubation based on physician decision rules are the major research and development in this study.

A learning classifier system (LCS), which is a rule-based machine learning method, interacts with the environment to generate an optimal policy, modifying the classifier parameter values according to the environmental feedback to formulate more effective rules. A popular type of LCS is the extended classifier system (XCS), for which numerous applications have been proposed [11,12]. The input data for the original XCS must be encoded into binary format. However, most problems are more easily addressed by encoding with real values. Therefore, Wilson modified the XCS for the adoption of real-value inputs; an XCS that can process such inputs is called an XCSR [13]. One feature of XCS is the “don’t care” bits in the rules. This feature enables users to determine which variables are less important in some situations but more important in other situations. A study applied this feature to identify the critical variables for detecting Internet addiction in patients [14]. Thus, the XCSR is suitable for the present objective: determining and evaluating SBT-related variables are more important for successful extubation than others. The aim of this study was not only to construct a model for predicting extubation outcomes but also to evaluate the critical SBT effect with other variables.

## 2. Experimental and Computational Details

### 2.1. Subjects Description

This retrospective study of the electronic medical records (EMR) database has been approved by the Institutional Review Board of Kaohsiung Medical University Chung-Ho Memorial Hospital (the approval number: KUMHIRB-F(1)-20200033). A total of 262 deidentified patient data were obtained from the above hospital. The recruited patient is on a ventilator for the first time and for more than 48 h and has stable vital signs before extubation. Since the literature has indicated that SBT with limited pressure support is more effective than T-tubes [15,16], the data of a 6 cmH_2_O pressure-supported SBT is collected in this study.

Figure 1 presents the extubation procedure used in the ICU of the internal medicine department. The intensivist decides to stop ventilation support and execute the extubation procedure after each index meting the relevant criteria. Within 48 h of extubation, the patient who experienced respiratory failure or died is defined as the extubation failure group, and the patient who did not is defined as the extubation success group. Descriptions of the variables obtained from the EMR are presented in the following section.

### 2.2. Variable

For evaluating critical factors, 40 variables for each patient were selected as the input variables. Table 1 lists the selected 19 variables of clinical information, consisting of basic patient information, two severity scales, mechanical ventilator-related information, hemodynamics situation before doing SBT, maximal inspiratory pressure, and disease history (Table 1). Table 2 presents another 21 variables of acquired information when doing SBT, including the hemodynamics and ventilation level at the 1st, 30th, 60th, and 90th minute. Regarding the output variable, the patient in the extubation failure group is encoded as 0 and the one in the success group is encoded as 1.

### 2.3. XCS and XCSR

The learning classification system (LCS) is a rule-based machine learning method that combines exploration components (evolutionary computation, EC) and learning components (reinforcement learning, RL). The development of LCS has been decades-long, and the most famous descendant in the field of the LCS is the extended classifier system (XCS). The XCS retains the main components of the LCS, but still achieves similar performance or even better. The streamlined architecture makes the XCS a benchmark for future development in the LCS field.

The architecture of the XCS is a standard RL. The XCS can be regarded as an agent interacting with the environment to achieve the desired goal. The environment can be a database to be explored or a problem to be solved. The XCS uses a set of cooperative rules to express the solutions it has learned. The rules are easily interpretable, and the expression of the rules is “IF condition THEN action”. In addition, each rule also contains 3 parameters to evaluate the quality of the rule: (1) prediction, predict the reward after the action is executed; (2) prediction error, calculate the error between predicted and actual reward; (3) fitness, evaluate the quality and reliability of rule. The reward received from the environment is used to update the rule parameters using Q-learning and the Widrow–Hoff delta rule. Introducing EC can help the XCS reorganize and explore better rules. The XCS has also been proven to be able to find the most general and accurate rules.

The operation process of the XCS is described as follows: (1) Environmental input is converted to binary encoding through the detector. The set of rules is called population [P] in the XCS. The condition part of the rule will correspond to the environmental input and use ternary-alphabet {0, 1, #} for encoding. The # symbol is treated as “don’t care”, which means it can match regardless of whether the corresponding environmental input is 0 or 1. The action part of the rule adopts binary encoding. The XCS will search for all the rules in [P] and move the rules whose condition part matches the environmental input to the matching set [M]. If [M] is empty, the XCS will start the covering mechanism. In the covering mechanism, the XCS will generate a new rule that matches the condition part with the environmental input and randomly give a binary code to the action part. In the condition part, there is a certain probability to replace the regular bit with the # symbol. The remaining parameters are given random initial values. The generated rule will be placed in [P] and rescan [P] to generate [M]; (2) The rules in [M] may contain different actions. To calculate the expected reward for each action, the rules in [M] with the same action are used to execute the fitness weighted average as the expected reward of the action. The prediction array [PA] can be built after calculating the expected reward of each action. The XCS has two modes to select action for execution, namely exploration and exploitation. In the exploration mode, the XCS adopts the roulette-wheel method, which means that the larger the expected reward, the easier to be picked. In the exploitation mode, the XCS directly selects the action with the most expected reward to execute. The selected action will be executed through the effector, and the environment will give reward. The rules in [M] whose action is the same as the selected action will be moved to action set [A]; (3) The reward obtained from the environment will be used to update the prediction, prediction error, and fitness parameters of rules in [A] through Q-learning and the Widrow–Hoff delta rule; (4) The parameters of the rule corresponding to [A] in [P] will be updated; (5) The EC method in the XCS is to use genetic algorithm (GA). When the average time of the rule in [A] from the last GA execution is greater than the threshold, GA will execute. GA will use fitness as the probability to select two rules for crossover and mutation to generate two new rules. In the XCS, the alleles of condition in the two rules will be exchanged with probability χ, and the alleles will be randomly flipped and mutated with the probability μ; (6) The XCS also has subsumption mechanism to assist in learning the most general and most accurate rules. The XCS will determine two rules corresponding to the same environmental input, and whether the more general rule has enough experience and reliability to accommodate another specific rule; (7) The XCS will repeat the above steps until it converges or reaches the maximum number of iterations.

The variables in real-world data are often continuous values. A new encoding is proposed in the condition part of the rule to allow the XCS to handle input data other than binary. The modified XCS version is called XCS with real-valued input (XCSR) [13]. The schematic of the XCSR and the corresponding operation procedure are shown in Figure 2 and Figure 3, respectively. The XCSR mainly converts the encoding of the condition of the rules from ternary-alphabet to interval predicates. The expression of interval predicate is int_i_ = (c_i_, s_i_), where int_i_ represents the i-th interval predicate corresponding to the i-th input variable, c_i_ represents the center, and s_i_ represents the spread. When an i-th input variable x_i_ can match the i-th interval predicate int_i_, it means (c_i_ − s_i_ ≤ x_i_ < c_i_ + s_i_). The operating mechanism of XCSR is the same as XCS, except that the mechanism is slightly changed in covering and GA. When covering generates a new rule, it will generate int_i_ corresponding to the x_i_ in the condition of the rule, where the value of c_i_ is set to x_i,_ and the value of s_i_ is set to a random number ranging from 0 to s_r_, where s_r_ is defined by the user. In the part of GA, the int_i_ of the two rules will be exchanged with probability *χ*. c_i_ and s_i_ will randomly add or subtract the value of m_i_ with probability μ, where m_i_ is defined by the user.

### 2.4. Data Analysis

This study will import clinical data of 228 patients with 40 variables as input data for the XCSR. The rule condition of the XCSR contains 40 interval predicates, and the output action is the success or failure of extubation. Table 3 shows the parameter settings of XCSR used in this study. The detailed description of XCSR can be referred to [13]. After the values of the 40 variables of the 228 patients were input into the XCSR, it generated 154 rules for extubation outcome prediction after the model reached stability. Therefore, according to the number of “don’t care” bits of each variable in the rules, the usage probability Prob_usage_ can be calculated. A larger Prob_usage_ in the *i*th variable indicates that more model rules use this variable to determine the extubation outcomes. The Prob_usage_ of the *i*th variable is calculated as follows:(1)$${\mathrm{Prob}}_{\mathrm{usage}}\left(i\right)=\frac{total\;rules\;number-don^{’}t\;care\;bit\;number\left(i\right)}{total\;rules\;number}\times 100\%$$

We constructed the model 10 times to verify the stability of the Prob_usage_ of each variable. Therefore, the results are presented as means ± standard deviations of Prob_usage_.

### 2.5. Statistical Analysis

Because the two groups were of unequal size, the nonparametric Mann–Whitney U test was performed to examine the distribution between extubation success and failure. The continuous variables are expressed as means ± standard deviations, and the discrete variables are expressed as numbers. A *p* value of <0.01 indicated a significant difference between two groups in two-tailed tests. The area under the receiver operating characteristic (ROC) curve (AUC) was used to test the discriminatory ability of each variable, with results expressed as the AUC values. The analyses were conducted and the XCSR was coded using the LabVIEW platform (National Instruments Corporation, Austin, TX, USA).

## 3. Results and Discussion

After exclusions were made for 34 of the 262 patients because of missing data, the data of 228 patients remained for analysis. Table 4 presents the clinical information of the final sample. Height and ideal body weight (IBW) differed significantly between the two groups. The AUC of these two variables were both 0.63.

Table 5 lists the variables measured during the SBTs of the 228 patients. A significant between-group difference was only observed for the heart rate (HR) in the 60th minute. The AUC of this variable was 0.63.

Figure 4 presents the performance of one of the XCSR models (a total of 10 models were built, one of them for an example). A problem refers to an iteration of a prediction that has been created. For the model to approach 100% accuracy, only approximately 125 problems were required. Though the possibility of over fitting of the XCSR model cannot be excluded, the analysis of the rules created by the XCSR can still be useful information for clinical doctors to make a decision.

Figure 5 presents the “don’t care” numbers of each variable in all rules after sorting. The first six variables had significantly fewer “don’t care” numbers, meaning that they had higher usage probabilities. Table 6 lists both these high-performing variables and some others with lower rankings.

Variables ranked 1–6 had a substantially higher Probusage than did other model variables. The Probusage of all variables ranked ≥7 was lower than 36%.

Although height, IBW, and HR in the 60th minute typically differ significantly between the extubation failure and success groups in relevant studies, the AUC values remain lower than 0.63. Few studies using the XCSR have applied the ROC analysis. However, in clinical research, ROC analysis is the most common method for comparing different models and variables. Therefore, in the present study, the rewards in the prediction array were applied to the XCSR model for the ROC analysis. The AUC of 0.993 was significantly higher than that of individual variables. Moreover, the accuracy of the XCSR model in predicting extubation outcomes approached 100% after training. For comparison with the XCSR model, the accuracy of height, IBW, and HR in the 60th minute was determined as 59%, 65%, and 65%, respectively, by using the cutoff point from the ROC analysis to predict extubation outcomes. The prediction accuracy for individual variables was relatively low because the ratio between the number of extubation failures and successes was approximately 0.83, meaning that the accuracy would be 83% if the model predicted that all extubation outcomes would be successful. Nevertheless, these results demonstrate how powerful the present XCSR model is. In the population rule set, 130 and 24 rules predicted extubation success and failure, respectively. Overall, the rules predicted the extubation outcomes with high accuracy.

After the model was constructed 10 times, although the Prob_usage_ rank of some variables differed, the first six positions remained filled by the same variables. Moreover, the Prob_usage_ of these six variables was higher than 95%, with small standard deviations. Overall, the results indicate that the respiratory rate (RR) and minute ventilation (VE) measured during the SBTs, as well as body weight (BW), were more important than the other variables. In particular, VE was critical at all-time points during the SBT. Studies have demonstrated that some SBT-related respiratory variables are risk factors for extubation failure [10,17,18,19]. One study noted that VE is a parameter that is easy to measure during SBTs that can guide decisions regarding the discontinuation of mechanical ventilation [17]. Another study reported that including all variables (especially the variables in 90th minutes) assessed during SBTs in a training model resulted in higher performance [20]. The current results also suggest that VE during SBT is a key predictor of successful extubation.

Patients who are able to complete an SBT are likely to have more stable breathing than those who are not. This may explain why respiratory rate in the 90th minute constitutes a major factor in extubation decision-making. Notably, BW can also be a critical factor for successful extubation. Because the rules generated by the XCSR are presented in a range, the values may not be continuous. However, the reason BW is typically used for rules in XCSR models remains unclear. We speculate that patients may feel more comfortable and breathe more stably if their BW is regulated within a certain range. Further research is warranted to confirm this supposition.

Some studies have also applied machine learning to predict extubation success or failure. Zeggwagh used multivariable logistic regression (MLR) and three variables (vital capacity, RSBI, and maximal expiratory pressure) to create a prediction model [21]. Mueller (2004) observed that a model created using an artificial neural network (ANN) outperformed one created using MLR [22] in predicting extubation outcomes in infants. Another study revealed that an ANN model more accurately predicted extubation failure [23]. Mueller (2013) tested five machine learning methods, namely MLR, ANNs, boosted decision trees, naïve Bayesian classifiers, and support vector machines, to predict extubation outcomes in a large sample of premature infants [24]. Two common statistical models, MLR and partial least squares (PLS) regression, were also used to test the same dataset of present study. The AUCs of MLR (0.660) and PLS (0.812) were significantly lower than that of the XCSR model (0.993). Table 7 presents a summary of the AUC results obtained in previous studies and the current study. Because each study applied different database and variables for model training, the summary of Table 7 is just for reference instead of comparison. Interpreting the nonlinear models and determining which variables were more important than others are challenging, especially when ANNs are used. Relevant studies have typically employed the trial-and-error approach to fine-tune models and variables. Though some method had been proposed for NN model interpreting, the XCSR model is easier to interpret. In the present study, each rule created by the XCSR was readable (each variable in a rule would be a range of value or “don’t care”). In general, interpreting an XCSR model is easier than a model formulated using other nonlinear machine learning methods. The usage probabilities calculated in the present study may be useful for identifying the importance of each variable. In addition, the distributions of each variable in different rules also can be useful information for finding the physiological meaning of patients with fail extubation and patients with successful extubation. These rules may serve as a valuable reference for further research.

Recently, a study also proposed a method to visualize rule-based machine learning [25]. In future work, it is possible to apply the method for identifying the importance of each variable to compare with present study and to explore detailed information from the XCSR. In addition, the original XCSR had been proposed for almost two decades. Many modified versions of the XCSR were proposed in these years. To study which one of the modified XCSRs is more suitable for the purpose of this study is also a future work.

The present XCSR system is already built in the system of Kaohsiung Medical University Chung-Ho Memorial Hospital for providing the extubation suggestion. In the future work, the prospective research of this XCSR system for extubation suggestion will be investigated.

## 4. Conclusions

The XCSR model trained in the present study had a prediction accuracy of almost 100%. We identified critical factors of extubation success and failure from among 40 variables, observing that the respiratory variables assessed during SBTs were more important. The generated rules used over 95% of the VE measured in the 1st, 30th, 60th, and 90th minutes, as well as the respiratory rate in the 90th minute and BW. These parameters, especially VE, are the key to determining extubation success or failure. These findings suggest that the duration of an SBT should be longer than 90 min, and that extubation decisions should be made in consideration of the variability in such respiratory parameters during an SBT. Overall, relative to those of other methods, the AUC of the XCSR model was higher, facilitating the evaluation of the importance of each variable. Furthermore, each generated rule may constitute a useful reference for future investigations.

## Figures and Tables

**Figure 1 bioengineering-09-00701-f001:**
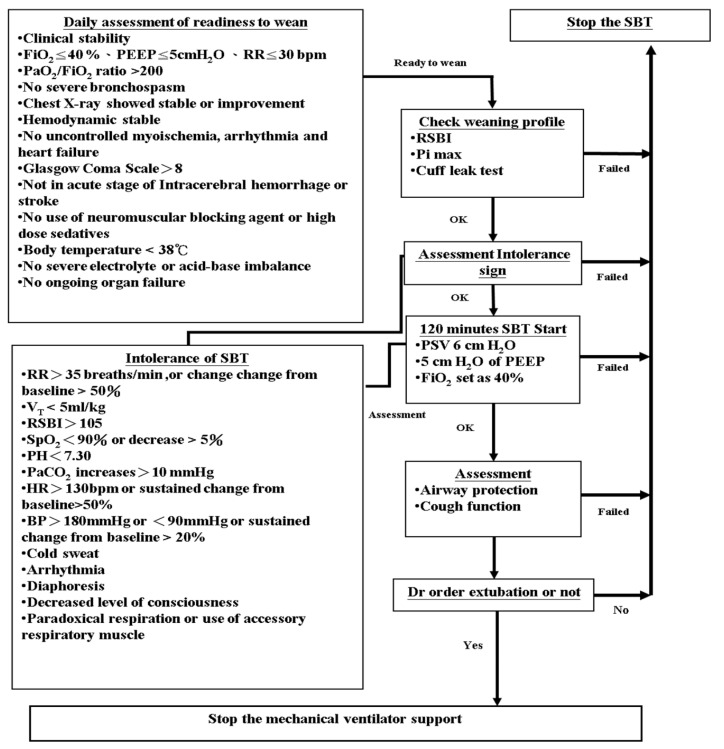
Ventilator weaning protocol in ICU of Kaohsiung Medical University Chung-Ho Memorial Hospital.

**Figure 2 bioengineering-09-00701-f002:**
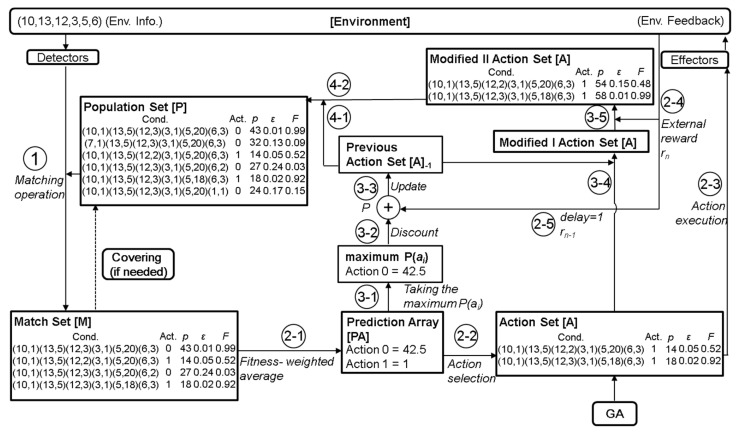
Schematic of the XCSR.

**Figure 3 bioengineering-09-00701-f003:**
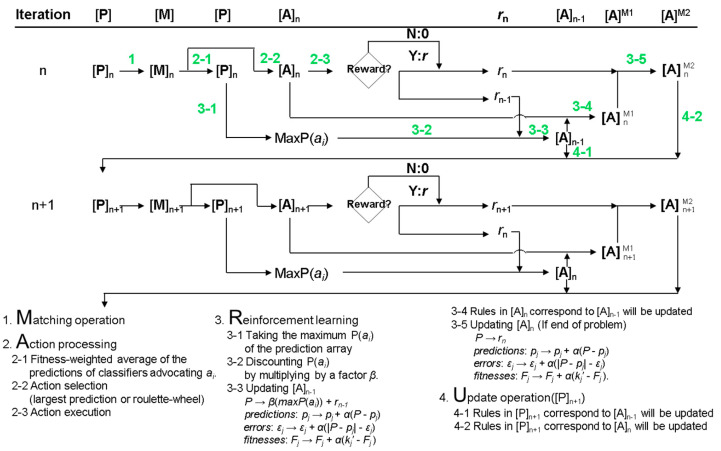
Flowchart of the XCSR in stage n.

**Figure 4 bioengineering-09-00701-f004:**
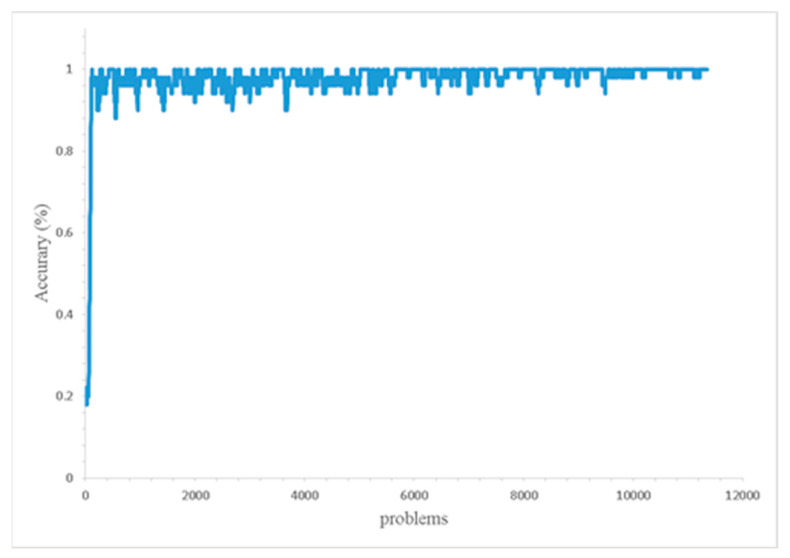
XCSR performance.

**Figure 5 bioengineering-09-00701-f005:**
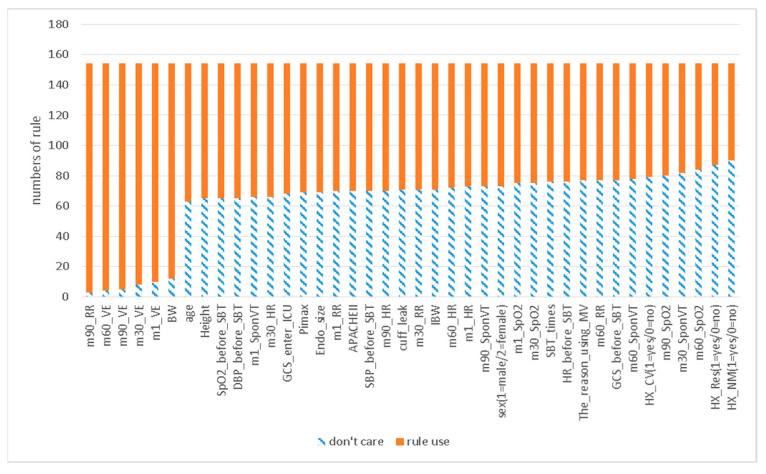
Distribution of the number of rules that use each variable.

**Table 1 bioengineering-09-00701-t001:** Selected clinical variables.

Variable (Unit)	Definition
**Basic information**	
Gender (a.u.)	Subject gender (1: male; 2: female)
Age (year)	Subject age
Height (cm)	Height
BW (kg)	Body weight
IBW (kg)	Ideal body weight
**Severity scale**	
APACHEII (a.u.)	Acute physiology and chronic health evaluation II
GCS_enter_ICU (a.u.)	Glasgow coma scale when entering ICU
GCS_before_SBT (a.u.)	Glasgow coma scale before doing SBT
**Mechanical ventilator**	
Endo_size (mm)	Endotracheal tube diameter
cuff_leak (mL)	Leak of cuff of endotracheal tube
Reason_using_MV (a.u.)	Reason of using mechanical ventilator
**Hemodynamics** **before SBT**	
SBP_before_SBT (mmHg)	Systolic blood pressure before SBT
DBP_before_SBT (mmHg)	Diastolic blood pressure before SBT
HR_before_SBT (bpm)	Heart rate before SBT
SpO_2__before_SBT (%)	Pulse oxygen saturation before SBT
**Ventilation**	
Pimax (cmH_2_O)	Maximal inspiratory pressure
**Disease history**	(0: no; 1: yes)
HX_Res (a.u.)	History of respiratory disease
HX_CV (a.u.)	History of cardiovascular disease
HX_NM (a.u.)	History of neuromuscular disease

**Table 2 bioengineering-09-00701-t002:** SBT-related cardiovascular and respiratory variables.

Variable	Definition
SBT_times (a.u.)	How many times of doing SBT
**Hemodynamics** **during SBT**	
m*X*_HR (bpm)	*X*th minute heart rate
m*X*_SpO_2_ (%)	*X*th minute pulse oxygen saturation
**Ventilation during SBT**	
m*X*_VE (mL)	Xth minute ventilation volume
m*X*_SponVT (mL)	Xth minute spontaneous tidal volume
m*X*_RR (bpm)	Xth minute respiratory rate

**Table 3 bioengineering-09-00701-t003:** Parameter setting.

Parameter	Value	Definition
*Epoch*	100	Number of dataset iterations
*N*	1500	Maximum number of rules in [P]
*β*	0.2	Learning rate for updating the parameters
*α*	0.1	Decline rate of fitness
*ν*	0.1	Exponent of fitness
*ε_0_*	10	Acceptable range of prediction error threshold
*θ_GA_*	25	The threshold for GA executing
*χ*	0.8	Probability of GA performing crossover
*μ*	0.04	Probability of GA performing mutation
*θ_del_*	20	The threshold for deleting rule
*δ*	0.1	The fraction of the mean fitness in [P]
*θ_sub_*	20	The threshold for subsumption
*P_#_*	0.5	When covering, the range of int_i_ can cover the maximum and minimum values of x_i_ with a certain probability
*p_i_*	10	Initial value of prediction
*ε_i_*	0	Initial value of prediction error
*F_i_*	10	Initial value of fitness
*θ_mna_*	2	Maximum number of actions

**Table 4 bioengineering-09-00701-t004:** Characteristics of the final sample of 228 patients.

Clinical Characteristics	Extubation Failure (*n* = 40)	Successful Extubation (*n* = 188)	Total (*n* = 228)	AUC
**Basic information**				
Gender (male/female)	18/22	118/70	136/92	-
Age (year)	72.62 ± 14.05	67.53 ± 15.76	68.42 ± 15.57	0.60
Height (cm)	158.5 ± 9.05	162.60 ± 08.56 *	161.89 ± 8.76	**0.63**
BW (kg)	60.73 ± 13.22	60.94 ± 14.54	60.90 ± 14.29	0.51
IBW (kg)	55.51 ± 6.36	58.27 ± 06.03 *	57.78 ± 6.16	**0.63**
**Severity scale**				
APACHEII	24.43 ± 8.19	21.29 ± 08.33	21.84 ± 8.38	0.61
GCS_enter_ICU	6.30 ± 3.41	6.84 ± 03.52	6.74 ± 3.28	0.55
GCS_before_SBT	9.55 ± 2.00	10.10 ± 01.63	10.00 ± 1.71	0.60
**Mechanical ventilator**				
Endo_size				
7.0 # (%)	9 (3.9)	31 (13.6)	40 (17.5)	-
7.5 # (%)	31 (13.6)	156 (68.4)	188 (82.5)	-
cuff_leak	203.47 ± 124.36	245.65 ± 107.31 *	283.25 ± 111.36	0.63
Reason_using_MV	21/19	93/95	114/114	-
**Hemodynamics before SBT**				
SBP_before_SBT	144.25 ± 26.34	140.94 ± 26.47	141.52 ± 26.42	0.53
DBP_before_SBT	71.40 ± 17.39	70.89 ± 15.86	70.98 ± 16.10	0.52
HR_before_SBT	92.95 ± 16.96	88.15 ± 15.91	88.99 ± 16.16	0.57
SpO_2__before_SBT	97.50 ± 2.18	98.22 ± 01.87	98.09 ± 1.95	0.60
**Ventilation**				
Pimax	33.25 ± 11.99	34.82 ± 12.19	34.55 ± 12.14	0.54
**Disease history**				
HX_Res (yes/no)	12/28	44/144	58/170	-
HX_CV (yes/no)	30/20	104/84	134/94	-
HX_NM (yes/no)	4/36	31/157	35/193	-

* *p* < 0.01 between extubation failure and successful extubation.

**Table 5 bioengineering-09-00701-t005:** SBT variables of the 228 patients.

Variables during SBT	Extubation Failure (*n* = 40)	Successful Extubation (*n* = 188)	Total (*n* = 228)	AUC
SBT_times	2.58 ± 1.41	2.49 ± 1.49	2.51 ± 1.56	0.54
**Hemodynamics**				
m1_HR	94.15 ± 16.28	89.07 ± 15.77	89.96 ± 15.94	0.59
m1_SpO_2_	97.70 ± 2.29	98.02 ± 2.13	97.96 ± 2.15	0.54
m30_HR	93.90 ± 15.97	88.69 ± 17.55	89.60 ± 17.36	0.60
m30_SpO_2_	97.40 ± 3.10	97.71 ± 2.46	97.65 ± 2.58	0.51
m60_HR	96.40 ± 17.04	**88.12 ± 20.33** *	89.57 ± 20.01	0.63
m60_SpO_2_	97.58 ± 2.77	96.12 ± 12.47	96.38 ± 11.39	0.52
m90_HR	92.17 ± 21.86	85.94 ± 20.36	89.04 ± 20.72	0.61
m90_SpO_2_	87.78 ± 29.73	92.12 ± 23.11	91.36 ± 24.38	0.53
**Ventilation**				
m1_VE	7.93 ± 2.97	7.94 ± 2.56	7.94 ± 2.63	0.52
m1_SponVT	420.40 ± 206.83	463.26 ± 222.54	455.74 ± 220.03	0.58
m1_RR	19.60 ± 6.89	18.73 ± 6.61	18.88 ± 6.66	0.53
m30_VE	7.53 ± 2.46	7.90 ± 2.78	7.83 ± 2.72	0.53
m30_SponVT	442.27 ± 244.51	531.98 ± 706.51	516.24 ± 650.11	0.60
m30_RR	18.65 ± 6.19	17.37 ± 5.68	17.60 ± 5.78	0.57
m60_VE	8.20 ± 3.46	7.84 ± 2.62	7.90 ± 2.78	0.52
m60_SponVT	451.43 ± 208.80	472.49 ± 174.87	468.79 ± 180.95	0.55
m60_RR	18.87 ± 6.22	17.57 ± 6.17	17.80 ± 6.19	0.57
m90_VE	7.65 ± 2.63	7.81 ± 2.45	7.82 ± 2.43	0.53
m90_SponVT	466.93 ± 191.02	478.64 ± 187.76	478.68 ± 185.68	0.55
m90_RR	17.65 ± 4.01	17.52 ± 6.12	17.62 ± 5.69	0.53

* *p* < 0.01 between extubation failure and successful extubation.

**Table 6 bioengineering-09-00701-t006:** Usage probabilities of the top 10 variables. The first 6 variables have significant higher probabilities.

Ranking	Variable	Prob_usage_ (%)
1	m90_RR	96.74 ± 1.27
2	m60_VE	96.58 ± 1.92
3	m90_VE	96.07 ± 1.67
4	m30_VE	95.90 ± 1.49
5	m1_VE	95.71 ± 1.54
6	BW	95.61 ± 2.70
7	age	36.10 ± 5.23
8	Height	35.29 ± 3.33
9	SpO_2__before_SBT	34.93 ± 4.77
10	DBP_before_SBT	34.76 ± 4.43
…	…	…
40	HX_NM	26.03 ± 5.28

**Table 7 bioengineering-09-00701-t007:** AUC results of the current and previous studies (bold text: the result of our group).

Ref.	Method	Type	Extubation Failure *n* (%)	AUC
Zeggwagh, et al. (1999) [21]	MLR	Calibration	22 (36.67)	0.913
	MLR	Validation	22 (37.29)	0.855
Mueller, et al. (2004) [22]	MLR	Calibration	22 (16.92)	0.810
	MLR	Validation	13 (24.53)	0.750
	ANN	Calibration	22 (16.92)	0.810
	ANN	Validation	13 (24.53)	0.870
Mueller, et al. (2013) [24]	MLR	Calibration	59 (12.14)	0.880
	MLR	Validation	59 (12.14)	0.762
	ANN	Calibration	59 (12.14)	0.930
	ANN	Validation	59 (12.14)	0.753
	BDT	Calibration	59 (12.14)	1.000
	BDT	Validation	59 (12.14)	0.513
	NBC	Calibration	59 (12.14)	0.610
	NBC	Validation	59 (12.14)	0.626
	SVM	Validation	59 (12.14)	0.493
Hsieh, et al. (2018) [23]	ANN	Calibration	185 (5.14)	0.850
Chung, et al. (2020) [20]	MLR	Calibration	**28 (16.57)**	**0.889**
This study (2022)	MLR	Calibration	**40 (17.54)**	**0.660**
	PLS	Calibration	**40 (17.54)**	**0.812**
	XCSR	Calibration	**40 (17.54)**	**0.993**

## Data Availability

The data presented in this study are available on request from the corresponding author. The data are not publicly available due to participates’ privacy.

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
