# Peer review of "Toward Evaluating Critical Factors of Extubation Outcome with XCSR-Generated Rules"

_bioengineering, 2022, doi:10.3390/bioengineering9110701_

Round 1
Reviewer 1 Report
Good research, but quality of figure is poor. This manuscript needs critical revision about Figure.
Author Response
Point 1: Good research, but quality of figure is poor. This manuscript needs critical revision about Figure.
Response 1: Thank you for your appreciation and suggestions. We have replaced Figures 2 and 3 with clearer versions.
Reviewer 2 Report
None
Author Response
Response: Thank you for taking the time to review our manuscript.
Reviewer 3 Report
an interesting original approach on an understudies subject: factors of extubation outcome with XCSR -generated rules
manuscript is of interest for the readers of bioengineering
introduction sets the scene quite well
Methods are extensive with 3 complex original figures
deficient the highlight of this manuscript
minor: The de-identified 89 patient data were obtained ...phrase needs additional details , please
results and the new approach are interesting
Discussion is to fastly constructed - additional comparing to the existing models is necessary, sorry
Only the first part of the Conclusion section is necessary. This section of conclusions needs to be reduced and balanced. The rest of it is for Discussion section, please.
Author Response
Point 1: an interesting original approach on an understudies subject: factors of extubation outcome with XCSR -generated rules
manuscript is of interest for the readers of bioengineering
introduction sets the scene quite well
Methods are extensive with 3 complex original figures
deficient the highlight of this manuscript
Response 1: Thank you for the appreciation.
minor:
Point 2: The de-identified 89 patient data were obtained ...phrase needs additional details , please
Response 2: Thanks for pointing out the issue. The detail of the demographic of our data were showed in Table 4. In addition, we have added the number of our data in the subject description section "Total 262 de-identified patient data were obtained from above hospital".
Point 3: results and the new approach are interesting
Response 3: Thanks for your appreciation.
Point 4: Discussion is to fastly constructed - additional comparing to the existing models is necessary, sorry
Response 4: Thanks for pointing out the issue. In this study, the multivariable logistic regression and partial least squares regression also applied to built the model of our data. The result were showed in Table 7. In addition, the Table 7. also compare the result of our model with other research.
Point 5: Only the first part of the Conclusion section is necessary. This section of conclusions needs to be reduced and balanced. The rest of it is for Discussion section, please.
Response 5: Thanks for the advice. We have moved the second paragraph of conclusion to the discussion.